# Reproductive Biology of Male European Hake (*Merluccius merluccius*) in Central Mediterranean Sea: An Overview from Macroscopic to Molecular Investigation

**DOI:** 10.3390/biology12040562

**Published:** 2023-04-07

**Authors:** Alessia Mascoli, Michela Candelma, Alberto Santojanni, Oliana Carnevali, Sabrina Colella

**Affiliations:** 1Laboratory of Developmental and Reproductive Biology, DiSVA, Università Politecnica delle Marche, 60131 Ancona, Italy; 2National Research Council (CNR), Institute of Marine Biological Resources and Biotechnologies (IRBIM), 60125 Ancona, Italy

**Keywords:** *Merluccius merluccius*, male reproductive pattern, histological analysis, somatic indexes, reproductive regulation, Central Mediterranean Sea

## Abstract

**Simple Summary:**

The vast majority of the world’s commercial fish stocks are considered overfished. Among demersal species, in the Mediterranean Sea, the European hake (*Merluccius merluccius*) is experiencing high fishing pressure, resulting in a critical overexploitation status. The reproductive pattern of the male sex is poorly investigated. In order to provide scientific advice for its sustainable management, this study gave, for the first time in the Central Mediterranean Sea, an overview of the reproductive biology of male European hakes, by using a multidisciplinary approach. The sex ratio throughout the three-year period and by length class showed that males dominated the population from February to June and up to 24 cm, respectively. The reproduction occurred all year round with a seasonal peak in spring/summer. The macroscopic and histological sizes at first maturity were 18.6 cm and 15.4 cm, respectively. The entire reproductive cycle was unequivocally identified by five histological developmental stages of the testis. Finally, the molecular approach allowed us to investigate endocrine regulation from the onset of puberty to the adult lifecycle.

**Abstract:**

The improvement of scientific knowledge about overexploited fishery resources allow us to provide scientific advice for their management and stock protection. By using a multidisciplinary approach, the aim of the study was to characterize, for the first time in the Central Mediterranean Sea (GSA 17), the reproductive biology of males of *M. merluccius*, currently highly exploited. A multi-year sampling from January 2017 to December 2019 was performed to exhaustively evaluate the sex ratio of the stock, while the 2018 annual sampling was selected to investigate the reproductive pattern of males. Individuals in spawning conditions were found every month, proving that *M. merluccius* is an asynchronous species, reproducing all year round, with a seasonal reproductive peak in spring/summer, as indicated by GSI. Five gonadal development stages were defined to fully describe the reproductive cycle of males. The macroscopic and histological L_50_, respectively 18.6 cm and 15.4 cm, were both below the Minimum Conservation Reference Size (MCRS). According to the mRNA levels, *fsh* and *lh* played a significant role during spermiation, whereas the *gnrhr2a* was involved at the beginning of sexual maturity. In the testis, *fshr* and *lhr* reached maximum expression levels before the spermiation. The hormonal stimuli of 11-ketotestosterone and its receptor were significantly higher when the specimen was in reproductive activity.

## 1. Introduction

The Mediterranean Sea offers a wide variety of species of commercial interest, showing high diversity in fishing patterns, with a vast geographical dispersion of landing sites and islands [1]. It is considered one of the most economically important seas in the world [2]. Unfortunately, several factors such as the overexploitation of fishery resources, more and more severe anthropic pressure and high demand for marine resources, bycatch, habitat loss and degradation, the introduction of alien species, eutrophication, pollution, climate change, and ghost fishing are threatening the marine diversity [3], resulting in a general decline of resources [4].

In 2017, among the FAO’s 16 Major Fishing Areas, the Mediterranean and Black Sea (Area 37 [5]) had the highest percentage (62.5%) of stocks fished at unsustainable levels [4]. Based on data collected up to 2016 in the same area (FAO37), 41 out of 47 examined stocks resulted out of safe biological limits [6].

This panorama impelled countries and international organizations to act; in particular, the General Fisheries Commission for the Mediterranean (GFCM) of the Food and Agriculture Organization of the United Nations (FAO) and all its members established a new strategy to improve scientific knowledge and data collection to facilitate the adoption of effective management measures [7]. According to their report, although 75% of fish stocks still remain subject to overfishing, this percentage fell by more than 10% in 2014–2018, demonstrating that the newly adopted strategy is successfully inducing a countertrend.

In particular, among demersal species, European hake (*Merluccius merluccius,* L. 1758), which is highly exploited [8], had shown a marked reversal trend: its critical exploitation ratio decreased by 39% since 2013 [7]. This species plays a fundamental role in Mediterranean landings and is the second most important demersal fish species in terms of abundance and economic value [9,10,11,12]. Italy is the country that mainly exploits this species; in particular, Italian bottom trawl catches of the Northern and Central Adriatic Sea represent around 75% of the total Adriatic catches. In this area, landings present a fluctuating trend all over the time series considered accounting for the highest value in 2006 and the lowest value in 2019 [13].

It is widely distributed and inhabits the waters of the Atlantic Ocean, the entire Mediterranean Sea, and the southern waters of the Black Sea [14]. The bathymetric distribution is between 25 and 1000 m deep, with the highest densities recorded between 100 and 300 m. It has been observed that colder and nutrient-rich waters, such as the Atlantic, favour the existence of larger individuals that mature later than in warmer waters, such as the Mediterranean Sea, and specifically in the Adriatic. In the Mediterranean Sea, the size does not exceed 80–90 cm. In the northern and central Adriatic, females are already predominant at length values between 30 and 33 cm and represent the entire population when they reach 38–40 cm [15].

European hake is a gonochoristic species, multiple “spawner” with external fertilization of the eggs, indeterminate fecundity, and pelagic and fractional deposition [16]. Reproduction takes place throughout the year, however, shows different intensities and reproductive peaks that vary depending on the area.

Several studies on the reproductive biology of European hake have been conducted in different areas [10,17,18,19,20,21], but generally on females, which define the spawning stock biomass (SSB).

In the light of above, the aim of the current study is to characterize, for the first time in the Mediterranean Sea, in the Northern and Central Adriatic Sea area (GSA 17 [22]) the reproductive biology of males of *M. merluccius*, providing in-depth knowledge in basic research and contributing to complete the picture of the species. To achieve this goal, a multidisciplinary approach was used, from a macroscopic to a molecular point of view. Reproductive parameters, such as sex ratio estimation, macroscopic maturity staging, and somatic indices were assessed to obtain an overview of the reproductive traits of the male European hake. The histological examination was performed in order to investigate the reproductive cycle of the male and to verify the validity of the reference scale in use [23,24] or possibly modify it ad hoc for the species. The histological analysis was also used to compare macroscopic and microscopic approaches, reliability, and accuracy, since the macroscopic method gives a rapid and cheaper evaluation of the gonad, but suffers from relatively high error rates in assigning the reproductive stage. Furthermore, macroscopic and histological size at first maturity were evaluated and compared.

The study was completed by deepening the understanding of the regulation of reproduction: gene expression profile in the pituitary and testis and 11-ketotestosterone levels in plasma provided knowledge about the recurrence of the reproductive cycle in adults and the onset of the critical phase of reproduction and puberty [25,26]. Both mechanisms are determined by the functional competence of the brain-pituitary-gonad (BPG) axis [27].

## 2. Materials and Methods

**Sampling.** Wild European hakes (*Merluccius merluccius*) were sampled from January 2017 to December 2019 (except August due to the fishing ban) in the Northern and Central Adriatic Sea (GSA 17) in order to investigate the reproductive biology of the species (Figure 1).

Our study was based on two sources of samples:-random specimens were collected each month from all three years by professional fishing trawlers on the landing port, then preserved on ice and transported to the laboratory for analysis. The analysis performed on these specimens was finalized for macroscopic and histological investigations (both sex for sex ratio calculation, only males for somatic indices evaluation, histological gonad assessment, and size at first maturity estimation).-three additional biological samplings were carried out by researchers onboard commercial fishing trawl vessels, during the daily fishing activity, in June, July, and September 2018, in order to dissect some male specimens selected from the catch, collect fresh tissues (pituitary and testis were preserved in RNA until later use) and drew blood immediately after death from the caudal peduncle area, as required by the protocols of hormonal and molecular analysis. The operator selected individuals in good condition and at different sizes, in order to obtain a total sample that favored the male sex and involved all length classes.

Sampling was carried out at a bathymetric layer of 0–200 m and accidental fishing under the commercial size (total length < 20 cm) was also included, to more comprehensively investigate the size at first maturity.

The sample collection was performed under the guidelines of the Data Collection Framework Regulation (EU Reg.199/2008), establishing a community system for the conservation and sustainable exploitation of fisheries resources under the Common Fisheries Policy (CFP). The procedures did not include any animal experimentation and ethics approval was therefore not necessary, in accordance with the Italian legislation (D.L. 4 of March 2014, n. 26, art. 2)

**Sex ratio.** Males were distinguished from females by abdominal incision and gonad macroscopic inspection. The sex ratio of the European hake population was estimated both as the ratio between the total number of males and the total number of females [28,29] and as the proportion of females to the total sample [30,31,32,33,34], computed with the following equations:(1)Sex ratio=Male (n):Female (n)
(2)$$\mathrm{Sex}\;\mathrm{ratio}=\frac{Female\;\left(n\right)}{\left(Female\;\left(n\right)+Male\;\left(n\right)\right)}$$

Female specimens were considered for the sex ratio evaluation, and all the subsequent analyses only concerned males.

**Somatic indices**. Samples collected onboard were not used for somatic indices estimation, as it was not possible to carry out precise weighing for the gonad. The whole commercial sample (N = 219) was taken into account, using the same animals both for GSI and HSI. The total length of male specimens (L_t_, cm) was determined from the snout to the end of the caudal fin with an accuracy of 0.1 cm. The total weight (W_t_, g) was measured to the nearest 0.1 g. Testis, liver, and viscera were removed from each animal, and gonad weight (W_gon_, g), liver weight (W_l_, g), and gutted weight (W_gut_, g) were measured with an accuracy of 0.001 g. Following the recommendation of Somarakis et al. [35], somatic indices based on gutted, rather than total fish weight, were not influenced by the amount of food in fish stomachs.

The monthly variation of the gonadosomatic index (GSI) and hepatosomatic index (HSI) were evaluated according to the following equations: (3)$$\mathrm{GSI}=\frac{Gonad\;Weight\;\left(g\right)}{Gutted\;Weight\;\left(g\right)}\times 100$$
(4)$$\mathrm{HSI}=\frac{Liver\;Weight\;\left(g\right)}{Gutted\;Weight\;\left(g\right)}\times 100$$

**Histology**. Male European hake, from different monthly sampling and with different sizes, were randomly chosen to evaluate the gonadal developmental stage, compare it with the macroscopic staging method, and validate the reference maturity scale generally used for teleost males. Testis were fixed in 10% neutral formalin buffered with saline phosphate buffer, stored overnight at 4 °C, repeatedly washed with water and PBS 0.1 M, and finally stored in 70% ethanol at 4 °C until analysis. Samples were dehydrated through increasing ethanol concentrations and xylene, embedded in paraffin, and cut at a thickness of 4 μm by microtome (model RM2125 RTS; Leica Biosystem, Wetzlar, Germany). Consecutive sections were stained by Mayer’s hematoxylin and eosin method and examined under a microscope (Axio Imager 2; Zeiss, Oberkochen, Germany) at different magnifications. An assessment of the histological reproductive stage of samples was performed according to the maturity scale of Brown-Peterson et al. [23] already adapted to the species *M. merluccius* by Candelma et al. [24].

**Size at first maturity (L_50_)**. The classification of the maturity stage for male European hake was based on macroscopic and microscopic assessment in order to determine the macroscopic and histological size at first maturity, respectively. L_50_ was estimated by the logistic function according to Prager et al. [36]:(5)$$p=\left[1+e-r\;\left(x-x_{50}\right)-1\right]$$
where *p* is the proportion of mature males for each class of length, *r* is a fitted parameter, *x* is the total length, and *x*_50_ is the total length at which 50% of the males are mature. Specimens in the *immature* histological stage were classified as immature, and specimens in the *developing*, *early spermiogenesis*, *late spermiogenesis*, and *regenerating* stages were classified as mature.

**Gene expression profile.** Male selected specimens collected during the three biological samplings by researchers onboard commercial fishing trawl vessels were used to investigate the expression levels of some pituitary and testis genes implicated in the regulation of reproduction. Analyzed genes, primers, and their annealing temperature are shown in Table 1.

*RNA extraction and cDNA synthesis.* Total RNA was extracted from testes and pituitary glands using RNAzol RT Reagent (Sigma-Aldrich, St. Louis, MO, USA), following the manufacturer’s protocol. All RNA samples were treated with DNase 1 (Sigma-Aldrich Co., LCC., St. Louis, MO, USA) to remove any traces of DNA. According to Candelma et al. [37], a total amount of 1 μg per testis sample and 0.2 μg per pituitary gland sample were used for cDNA synthesis, employing the High-Capacity cDNA Reverse Transcription Kit (ThermoFisher Scientific, Waltham, MA, USA) and iQ5 iCycler thermal cycler (Bio-Rad).

*Real-Time PCR*. All qPCR assays were run in duplicate for each sample (both testis and pituitary gland), using the CFX Connect Real-Time System thermal cycler (Bio-Rad, San Diego, CA, USA). For each reaction, 1 µL of diluted (1/10) cDNA was combined with 0.2 µM forward primer, 0.2 µM of reverse primer, and 5 µL of 2X concentrated FluoCycle II SYBR Master Mix (Euroclone, Milan, Italy), containing SYBR Green as a fluorescent intercalating reagent. The thermal profile for all reactions was 3 min at 95 °C, 45 cycles for 10 s at 95 °C, 20 s at primer annealing temperature, and 20 s at 72 °C. The reference genes used to normalize data from cDNA were *beta-actin* (*b-act*) and *18S*. No amplification product was observed in non-template controls and no primer-dimer formations were observed in the control samples. 

The data obtained were analyzed using the iQ5 optical system software version 2.0 (Bio-Rad), including GeneEx Macro iQ5 Conversion and GeneEx Macro iQ5 files.


**Steroid Immunoassay**


The 11-ketotestosterone (11-KT) content in the plasma of male European hake was evaluated by a conventional enzyme immunoassay (EIA) developed for the Siberian sturgeon [38] and modified for its use in the sea bass [39].

Plasma from selected male specimens collected during the three biological samplings by researchers onboard commercial fishing trawl vessels were briefly extracted with methanol. The organic solvent was evaporated at 37 °C and the dry extract, visible as a pellet, was reconstituted by vortexing in a volume of assay buffer (EIA buffer, Cayman Chemical, Ann Arbor, MI, USA) equal to twice the initial volume of plasma. The assay was carried out in a final volume of 150 µL in 96-well microtiter plates coated with mouse anti-rabbit IgG monoclonal antibodies (Clone RG-16, Sigma-Aldrich, Inc., St. Louis, MO, USA); it was performed by using an 11-KT acetylcholinesterase conjugate (11-KT-AChE, Cayman Chemical, Ann Arbor, MI, USA) as tracer (0.1042 UE/mL), specific anti-11-KT rabbit antiserum [39] (diluted to 1:200,000), 11-KT standards (ranging from 1.0 ng/mL to 0.0005 ng/mL), or samples (50 µL). Plates were incubated overnight at 4 °C, under shaking conditions (140 rpm), rinsed, and color development was performed by adding 200 µL/well of diluted (1:50) Ellmans’s reagent followed by incubation under shaking conditions (140 rpm) for 4 h at 20 °C in the dark. Optical density was detected at 405 nm in a microplate reader (Bio-Rad microplate reader model 3550). The sensitivity of the assay was 0.002 ng/mL (Bi/B0 = 90%) and half-displacement (Bi/B0 = 50%) occurred around 0.015 ng/mL.

**Data Analysis.** All the statistical analyses were performed in the R environment, using R software version 3.6.2 (R Core Team, 2020).

The Chi-square goodness of fit test was adopted to determine whether the sex ratio monthly variation differed from the expected value (1:1). The null hypothesis (no difference between the observed and expected proportions) was tested at *p* < 0.05.

Statistical differences in somatic indices variations, gene expression profile, and 11-ketotestorerone plasma levels were checked by one-way analysis of variance (ANOVA) followed by a post hoc Tukey’s multiple comparison test. The confidence interval was set at 95% (*p* < 0.05) and results were expressed as *mean value ± standard error of the mean* (SEM) for somatic indices, or *mean value ± standard deviation* (s.d.) for gene expression and 11-KT plasma levels.

The calculation of the histological sample size was performed by Cochran’s Formula, in order to ensure reliable results for the gonad maturity stage assessment in the population.

A contingency table was used to compare the macroscopic staging method used during the commercial surveys with the histological staging method used in the laboratory. The percentage of agreement was calculated by dividing the total number of samples for which the methods agreed by the total number of specimens sampled during 2018. Cohen’s k coefficient [40] was applied to assess the agreement between the five stages according to the histological classification, considered as the reference baseline, and the five stages according to the macroscopic classification.

The statistical significance (*p* < 0.05) of the estimated parameters for macroscopic and histological L_50_ values was tested by the Wald test, the goodness of fit was assessed using the McFadden pseudo-R square (R^2^_MF_) [41] and the Cohen’s k coefficient [40] was applied to assess the agreement between the histological estimation, considered as the reference baseline, and the macroscopic estimation of L_50_. The likelihood ratio test was performed to compare the L_50_ values obtained by the macroscopic approach and the histology-based method.

## 3. Results

European hake specimens were monthly collected in the GSA 17 from January 2017 to December 2019 by professional fishing trawlers on a landing port. A multi-year sampling was available as the Institute for Marine Biological Resources and Biotechnology of the National Research Council (CNR-IRBIM) of Ancona is involved in planned research activities requested by the “Data Collection Framework-Biological Sampling of Commercial Catches” (DCF, EU Regulation 2017/1004), since 2006.

A total of 1316 collected samples were used to exhaustively estimate the sex ratio of the stock (females/males + females): 598 were males (45.44%) and 718 females (54.33%), according to the macroscopic evaluation of the gonad. Therefore, in the population of European hake during the three-year period, the sex ratio was biased towards females, significantly differing from the expected value of 0.5 (male:female = 1:1), as indicated by the Chi-square test of goodness of fit (sex ratio = 0.545; male:female = 0.83:1; χ^2^ = 10.942, df = 1, *p*-value = 0.00094; Table 2).

The monthly analysis showed an average value of 0.547 ± 0.039 (mean ± SEM) and the trend is reported in Figure 2: males slightly dominated the population from February to June, while females did so during the second half of the year. The Chi-square test of goodness of fit performed by months revealed that the sex ratio was significantly biased towards males in April and June, and towards females in November, December, and January (Table 2).

Figure 3 shows the gender distribution by length class: in the range of 14–24 cm, the *M. merluccius* population was dominated by males. At smaller sizes, only males were found, whilst starting from 25 cm, females became prevalent to make up 100% of the population from 31 cm onwards.

In order to highlight the reproductive pattern of males and perform an in-depth study, an annual smaller sample was selected from the total multi-year one, ensuring a homogeneous and reliable sub-population. Male specimens collected from January to December 2018 turned out to be a sufficient number to guarantee a dependable representation of the whole population, as shown by Cochran’s formula modified in the case of a smaller population (ideal sample size = 234, from the total N = 598). A total of 318 *M. merluccius* males, ranging from 13 cm to 29 cm, were sampled in 2018: 249 individuals were collected from professional fishing trawlers on landing ports, and 69 individuals directly onboard commercial vessels by researchers. The operator-on-board samplings were carried out excluding females and selecting males since the aim was to collect fresh samples from males to perform molecular and hormonal analysis.

Of the 249 samples collected from the landing port, 34 were excluded due to the poor condition of the gonad, which did not allow for accurate assessment, reducing the commercial sample size to N = 215 and the total one to N = 284 available for the further analysis described in the present work.

### 3.1. Somatic Indices

Samples collected on board were not used for somatic indices estimation, as it was not possible to carry out precise weighing for the gonad, and the whole commercial sample (N = 215) was taken into account, using the same animals both for GSI and HSI. The monthly variation of indices throughout 2018 is shown in Figure 4.

GSI mean values gradually increased from January to July, peaking between March and July, and remarkably decreased in September, keeping minimum levels until December. The hepatosomatic index (HSI) approximately kept constant levels throughout the year, with maximum values in June and November. All statistically significant differences detected in the GSI and HSI monthly values, evaluated by the one-way ANOVA and the post hoc Tukey’s test, are shown in Figure 4.

### 3.2. Histology

A total of 284 gonads were analyzed and 232 of them were histologically confirmed to be testis at different maturity stages. The microscopic aspect of the 232 testes revealed that the European hake has an unrestricted tubular testis. The reference scale^,^ from Brown-Peterson et al. [23] and Candelma et al. [24] was used as a guideline, and five stages were distinguished in the present work by specific macroscopic and histological features: *immature*, *developing*, *early spermiogenesis*, *late spermiogenesis,* and *regenerating*. In *immature* testis only primary spermatogonia are present and there is no lumen in lobules (Figure 5 A,B). The *developing* stage displays evident spermatocytes along lobules that can contain secondary spermatogonia, primary and secondary spermatocytes, spermatids, and rarely, spermatozoa still not released (Figure 5C,D). In *early spermiogenesis* testis, all germ cell stages can be detected, with the dominance of spermatocytes and spermatids. Small clusters of spermatozoa are found in the lumen of lobules and/or sperm ducts, after breaks of spermatocytes (Figure 5E,F).

*Late spermiogenesis* stage is characterized by a more considerable number of spermatozoa in the lumen of lobules and sperm ducts, compared to other stages (Figure 6A–D). In the stage *regenerating*, spermatocytes consist of primary and secondary spermatocytes, spermatogonia proliferate throughout the testis, the lumen of lobules and sperm ducts contain residual spermatozoa, and the discontinuous germinative epithelium is evident (Figure 6E,F). Individuals staged in *regenerating* showed some variability in characteristics, based on the time since the last spawning.

The monthly distribution of maturity stages is shown in Figure 7. Not all stages were found during monthly samplings. *Immature* individuals were sampled between June and October, while the other stages had a larger distribution throughout the year. Specimens in *late spermiogenesis* were found every month, from January to December, representing the most abundant stage until July and decreasing in autumn and winter.

The similarity percentage between the staging by the two different approaches (macroscopic and histological) was 18.1%. Considering the histological classification as the reference baseline, the percentage changed according to the maturity stage: the highest similarity percentage was registered for the *late spermiogenesis* stage (93.3%), conversely, the lowest values were recorded for *immature* (17.9%) and *developing* (0%) stages. In the cases of *early spermiogenesis* and *regressing*, the similarity percentage was 3% and 20%, respectively. Cohen’s k was 0.02 (95% confidence interval: −0.02–0.06), which corresponds to a “slight” level of agreement [42].

### 3.3. Size at First Maturity (L_50_)

The L_50_ was estimated by using all the specimens for which both macroscopic and histologic assessments of gonads were performed (N = 232). The size at first maturity based on the macroscopic staging of the testis was found to be 18.6 cm (Figure 8A). The estimated parameters of the logistic regression are summarized in Table 3, showing statistical significance (*p* < 0.05). The shortest size at which male specimens reached sexual maturity was 17 cm. The value of R^2^_MF_, which indicates the goodness of fit, was 0.71 (Table 4).

The same samples previously classified through a macroscopic analysis were validated by the histological approach and the logistic curve was computed (Figure 8B). The size at first maturity based on histological classification was found to be 15.4 cm. The estimated parameters of the logistic regression, summarized in Table 3, were statistically significant (*p* < 0.05), whereas the R^2^_MF_ was 0.57 (Table 4). According to the histological analysis, the shortest length at which male specimens reached sexual maturity, starting spermatogenesis, was 14.5 cm.

The macroscopic L_50_ and the histological L_50_ were significantly different (*p* < 0.05), according to the Likelihood ratio test (Table 4), the agreement between the two values was 76.7 % and the Cohen’s k was 0.24 (95% confidence interval: 0.13–0.36) (Table 4), which corresponds to a “fair” level of agreement [42].

### 3.4. Gene Expression Profile

From the total of 69 available samples collected directly onboard, the expression of the selected genes was evaluated only for those samples for which it was possible to obtain both a good RNA extraction, a consistent quantification of expression, and a reliable histological analysis to classify specimens according to the maturity stage of the gonad.

#### 3.4.1. Pituitary Gland

The mRNA expression of the gonadotropin-releasing hormone receptor 2a (*gnrhr2a*) showed the lowest value in the pituitary from *immature* specimens, gradually increased in *developing* until reaching the peak in *early spermiogenesis*, which was significant only compared to the *immature* stage (*p* < 0.05). The levels did not significantly decrease in *late spermiogenesis* (Figure 9A). Gonadotropin (*fsh* and *lh*) expression levels were also checked (Figure 9B,C). *fsh* gradually increased from *immature* to *late spermiogenesis*, where the peak was recorded as significant only compared to *immature* (*p* < 0.05). The trend of *lh* showed similar expression levels in *immature*, *developing*, and *early spermiogenesis* and sharply raised in *late spermiogenesis*, peaking significantly compared to the other stages (*p* < 0.05).

#### 3.4.2. Testis

Transcripts of receptors for Fsh (*fshr*), Lh (*lhr*), and androgens (*ar alpha*) were detected in the testis. *Fshr* had conspicuous expression from *immature* to *early spermiogenesis*, recording statistical significance in this stage, compared to *late spermiogenesis* (*p* < 0.05), where the expression remarkably decreased (Figure 10A). *lhr* showed the peak of expression in the *developing* stage, significantly different compared to the sudden decline both in *early* and *late spermiogenesis* testis (*p* < 0.05) (Figure 10B).

The trend of *ar alpha* showed low levels in *immature*, *developing*, and *early spermiogenesis* testis compared to the significant peak in *late spermiogenesis* one (*p* < 0.05) (Figure 10C).

### 3.5. 11-Ketotestosterone Immunoassay

From the total of 69 available samples collected directly onboard, the plasma levels of 11-ketotestosterone (11-KT) were evaluated only for those samples for which it was possible to obtain both a good steroids extraction, a consistent quantification by ELISA, and a reliable histological analysis to classify specimens according to the maturity stage of the gonad.

Figure 11 shows that specimens in the *immature* stage had 11-KT values close to zero; the levels increased in the other stages, with similar values between *developing* and *early spermiogenesis* and maximum registered in *late spermiogenesis*, which is significantly different compared to all the other stages.

## 4. Discussion

The overexploitation and the general decline of fishery resources caused by ever-increasing food demand and by environmental degradation become worrisome and make it urgent to take on management measures, supported by continuously updated scientific knowledge. Due to the economic importance and the overfishing status of the stock [17], the present study focused on the European hake (*M. merluccius*), providing for the first time in the Mediterranean, an exhaustive investigation of the reproduction of males using a multidisciplinary approach.

A first overview of the *M. merluccius* stock in GSA 17 was given by the sex ratio analysis. A multi-year sampling (2017–2019) provided an exhaustive picture of the state of the population. The sex ratio was slightly biased toward males from February to June and in shorter total length specimens, in accordance with previous studies in the Mediterranean Sea [19,43].

The spatial distribution of the population can be influenced by environmental factors such as oceanographic features, temperature, and food availability [44,45,46], but no information about possible sex-related differences in migration is available in the literature. The total sex ratio biased toward females is caused by the clear prevalence of females from October to January, while in the spawning season, this sex is outnumbered by males.

According to Soykan et al. [43], females generally dominate the stock, and males’ abundance declines after a certain size (>30 cm). Starting from 38–40 cm, in the Northern and Central Adriatic Sea, the hake population is totally composed of females [47] and it could be explained by the fact that the growth rate of European hake can be considered similar between sexes during the first year of their life, i.e., up to 20 cm in the Mediterranean area, as asserted by several studies focused on otolith microstructure analysis of juveniles. In the Catalan Sea, seasonal growth rates yielded an approximated length of 20 cm at the end of the first year [48]; this length was estimated to be 16 cm in the central Adriatic [49], 17 cm in the Aegean Sea [50], and 18 cm in the Tyrrhenian Sea [51]. After the first year of their life, around about 20 cm, males slow down their growth [49]. These studies supported our results, explaining why, in the sex ratio by length, males dominate the population up to 25 cm, then their presence decreased until the females make it up completely, starting from 31 cm.

The absence of differentiated females in the smallest class (13–14 cm) may be due to a different development between sexes; females mature when they reach a bigger size compared to males [20,21], and in such small sizes specimens could present a still undifferentiated gonad that could convert in the ovary later.

*M. merluccius* is considered an asynchronous species, with a protracted spawning season [52,53], reproductive all year round in the Atlantic Ocean and Mediterranean Sea, since mature specimens were frequent throughout the year [54,55]. In general, two spawning peaks were recognized: the first one occurs in winter, in deeper water, then adults move to shallower waters, during the reproductive season, recording the second peak in spring-summer [21,55,56,57]. However, depending on the geographical area, the spawning pattern could change and only one peak [10,53,54,58,59] or more than two could be recognized [60,61].

In the present study, the highest values of gonadosomatic index (GSI) were registered between March and July, indicating that there is only one peak in the reproductive cycle of males in GSA 17 during 2018 and it occurs in spring-summer. This result agrees with our previous works [20,37], where the reproductive pattern of *M. merluccius* females was investigated in the same geographic area and the GSI levels significantly increased in summer (June and April–July, respectively). The absence of the winter peak may be caused by the reduced bathymetry of the sampling carried out in the Adriatic Sea in the period in which this species migrates to deeper waters [56,62]. Moreover, variation in the spawning pattern of European hake could be merely attributable to regional discrepancies [43] or to a natural certain interannual variability [43]. According to Recanses et al. [10], the influence of temperature does not seem to be a key factor for the determination of spawning peaks of this species, but we do not exclude that in some areas with specific oceanographic conditions, it may be crucial. Furthermore, the annual evolution of the somatic indices is not as clear as in species with a shorter spawning season [43].

The reproductive activity requires energy, partly coming from ingested food, mainly from reserves in the liver and muscles. Therefore, it is reasonable to expect that the weight of the liver and muscle would reflect the accumulation and utilization of these energy reserves [63]. In the European hake, lipids are stored mainly in the liver, confirming the important role of this organ for energy storage [64]. In general, species adopt different reproductive strategies according to their dependence on energy storage: capital breeders build up reserves while resources are available and reproduce at a later time independently of food availability; income breeders allocate ingested food directly to reproduction [65]; the individual can also adopt a mixed strategy with income and capital co-occurring [66]. In the present study, the HSI trend was the same as that shown by females from the same study area [20]. There is no clear complementary pattern between GSI and HSI, suggesting that in this area (GSA 17), European hake can be defined as both income and a capital breeder, as argued by Carbonara et al. [21], in GSA 10, 18, and 19. Hake is such a plastic species that it can adapt its breeding strategy to the particular biotic and abiotic factors that characterize the geographical area [67]. In the Mediterranean Sea, considered a poorer sea compared to the Atlantic Ocean, during summer when many more resources are available, hake changes its reproductive strategy and behaves like an income breeder, i.e., the energy allocated to reproduction comes from concurrent feeding [67] and supports a long reproductive cycle. It could also be probable that this species feeds during the entire spawning period, as supposed by other authors in different Mediterranean areas [21,68].

Although biometrics, sex ratio, macroscopic classification of the gonads, and somatic indices trend estimation are the easiest, most rapid, and cheapest method to assess stock and fish reproductive status [69], it often lacks accuracy and precision [70]. Therefore, the histological approach allowed us to assign the stage of maturity to the gonad in an unambiguous and unequivocal way [71] and compare the two investigation methods.

For the first time in the Mediterranean geographic area, a deep investigation on the gonadal development of *M. merluccius* male throughout the life cycle was obtained by using a histological approach. The reference scale [23], already adapted to the species *M. merluccius* by Candelma et al. [24], did not turn out to be completely suitable for describing the entire reproductive cycle of males, as the characteristics shown in this study by the testis at different times of the year were not perfectly faithful to the reference description. For this reason, the existing reference scale was modified ad hoc, and five stages were recognized and distinguished by specific macroscopic and histological features. The stages in this work are defined as *immature* and *developing* perfectly coincide with the *immature* and *developing* described by Brown-Peterson et al. [23]. For the other stages, some differences were recorded and the *regressing* stage was not recognized, because males can reproduce throughout the year. There is no resting time, the latency period is so short that the last phase of one reproductive cycle coincides with the first phase of the next cycle, as other authors asserted in previous works [72]. Further confirmation is given by the presence of reproductive fish, specifically *late spermiogenesis* specimens, both before and after the reproductive peak (March–July), proving that for male European hake, the spermiation occurs throughout the year.

Histological validation is strongly recommended to ensure a proper calibration between visual and microscopic staging [73,74,75]. In this study, it is proved that European hake males’ maturity staging necessarily requires histological analysis and that the macroscopic classification causes many evaluation errors in all stages. Indeed, the similarity percentage between histological and macroscopic approaches was 18.1% and the Cohen’s k corresponded to a “slight” level of agreement [42]. Moreover, the highest level of discrepancy was recorded for the first two developmental stages, and it could be explained by the fact that small size testis displayed the typical macroscopic appearance of the *immature* stage (small, clear, threadlike testis), but in the histological investigation they showed advanced maturity stages. 

The assessment of the maturity stages of the gonad is a crucial step for the estimation of L_50_. Fish population dynamics studies required several parameters, such as the size at first maturity, which allows for the management and protection of stocks, defining the minimum catch size. Little is known about the size at first maturity for *M. merluccius* males: in the Atlantic Ocean, the estimated L_50_ is 32.8 cm [54] and 28.6 cm [55], in the Sea of Marmara and Aegean Sea L_50_ is 22.5 and 25.6 cm, respectively [43]. In the Adriatic Sea, values fluctuate in a range of 20–28 cm [9,22]. The present study confirms that in the Adriatic Sea, the European hake reaches maturity earlier than the other areas and therefore a smaller size. Indeed, the macroscopic L_50_ was estimated to be 18.6 cm, differently the histological L_50_ estimated was 15.4 cm. This result is related to an incorrect classification of macroscopically immature individuals, as confirmed by the 76.7% percentage of agreement and the “fair” level of agreement expressed by Cohen’s k [42]. Unlike European hake females [20,21], males are protected by Annex IX of the Council Regulation (EC) No 2019/1241 of 20th of June 2019 [76], in which the Minimum Conservation Reference Size (MCRS) is 20 cm. However, the protection of these species is not guaranteed because the females are still immature at 20 cm, the fishing activity could lead to a decrease in this resource with the collapse of the stock in upcoming years. Accordingly, the ability of exploited species to survive annual environmental variation can be negatively affected by the reduction of age and the average body size of the stock [77].

A series of internal and external factors stimulate and/or modulate the activation of the brain-pituitary-gonad (BPG) axis [27], a complex physiological mechanism of regulation that determines the ability to reproduce for the first time, or to enter puberty [25], once fish reach a certain size, and the recurrence of the reproductive cycle during the life, once fish became adults.

The present study investigated the role of the pituitary and gonad in the molecular regulation of reproduction. The *gnrhr2a* gene expression levels in the pituitary revealed an increase during the progress of the reproductive cycle peaking in the *early spermiogenesis* stage. This result suggests the involvement of the gene in the control of the early phase of sexual maturation, playing a key role in the reproductive cycle, as previously evidenced in seabass females [78]. The regulatory mechanism of gonadotropins is still unclear in asynchronous species. The *fshb* gene gradually increased from the *immature* stage, acting from the beginning of the cycle, and reaching the peak in the *late spermiogenesis* stage. This result can suggest that this hormone plays an ongoing role, probably because gonads contain all cell stages at the same time [24,79]. *lhb* reached the maximum level in the *late spermiogenesis* stage as *fshb*, but it probably acts subsequently, since in the previous stages the values are low. Overall, the gonadotropin trends are similar to other studies performed on multiple spawners [80,81,82]. In the testis, *fshr* and *lhr* showed earlier expression with respect to their ligands, a different kinetic of translation between mRNA codifying for the gonadotropins and their receptor may be hypothesized to guarantee the presence of the receptor upon arrival of the gonadotropins. Gonadotropins control steroidogenesis thus through androgens, whose mechanism of action is still an enigma in the reproduction of fish males, when sperm production occurs. In this study, the *ar alpha* receptor showed the highest expression level in the *late spermiogenesis* stage, that is when the animal is in full reproductive activity.

This result is supported by the profile of 11-ketotestosterone, the androgenic hormone that plays a key function in the endocrine control of male reproduction. A significant increase in this hormone in the *late spermiogenesis* stage explained the peak of *ar alpha* mRNA: in the last reproductive stage the testis is provided with the maximum availability of receptors in order to better respond to the paracrine hormonal stimuli of 11-KT.

## 5. Conclusions

The overexploitation status of the European hake stock makes it necessary to deepen the knowledge of the species, in particular focusing on reproductive biology. Unlike females, the male counterpart of *M. merluccius* stock in the Central-North Adriatic Sea is still protected by the Minimum Conservation Reference Size (20 cm), but the closed season [82] does not preserve the stock, since it is after the seasonal reproductive peak (March-July). The rapid and inexpensive overview of the reproductive cycle given by the macroscopic evaluation of gonad is not sufficiently appropriate for this species and the histological approach is essential to avoid bias in maturity stage assessment. Finally, the molecular and hormonal investigation can support the analysis of somatic indices and the macroscopic and histological L_50_ estimations to complete the scenario on the reproductive physiology of *M. merluccius* male. In conclusion, the multidisciplinary approach provided a deep knowledge of the reproductive biology of male European hake, furnishing further scientific information required to protect the stock and guarantee sustainable exploitation of this fishery resource.

## Figures and Tables

**Figure 1 biology-12-00562-f001:**
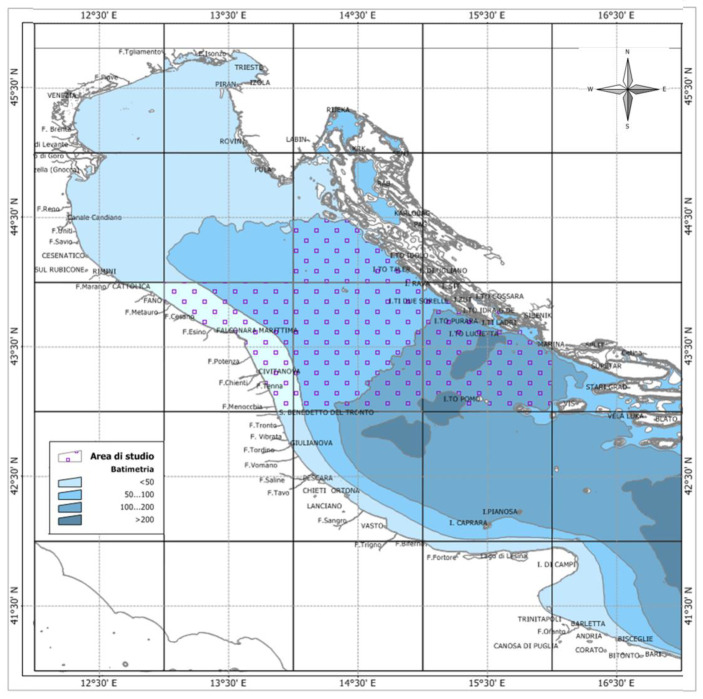
Map of the study area elaborated by Michela Martinelli, using the Manifold System 8.0 Universal Edition and list of the monthly sampling sites.

**Figure 2 biology-12-00562-f002:**
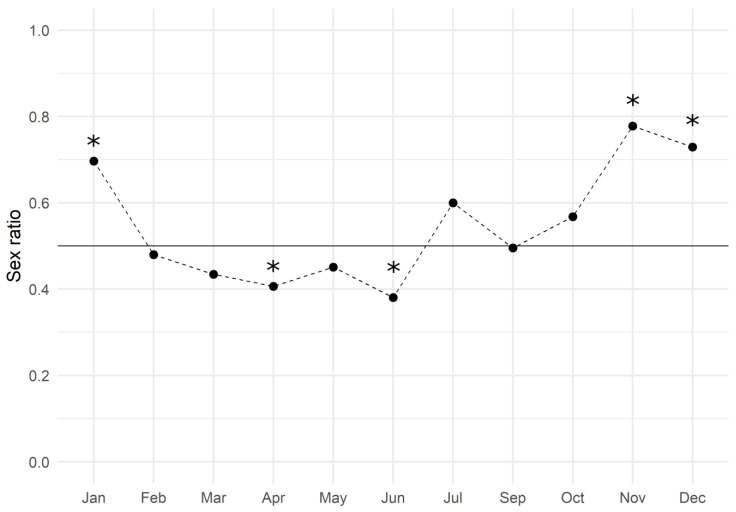
European hake sex ratio values throughout the three-year period of 2017–2019. The black line represents the expected gender distribution value of 1:1 (sex ratio = 0.5). The sex ratio was estimated by dividing the number of females by the total number of females and males (*females*/*males + females*). Asterisks indicate a significant difference (*p* < 0.05) in the monthly sex ratio compared to the expected value.

**Figure 3 biology-12-00562-f003:**
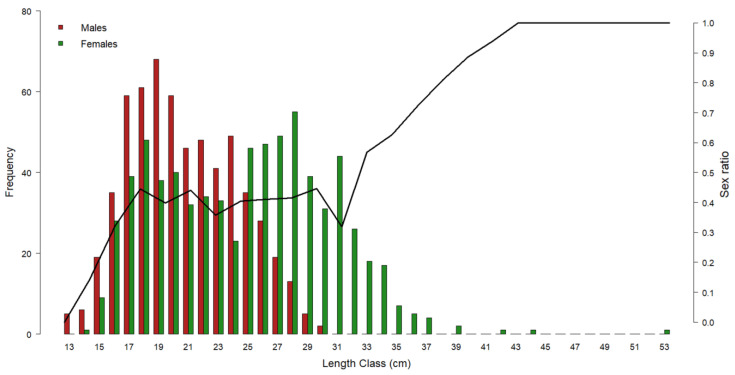
Representation of European hake gender frequency by length class throughout the three-year period of 2017–2019. Red bars = male frequency; green bars = female frequency. Black line = sex ratio by length class. Left y-axis refers to the histogram bars; right y-axis refers to the sex ratio values.

**Figure 4 biology-12-00562-f004:**
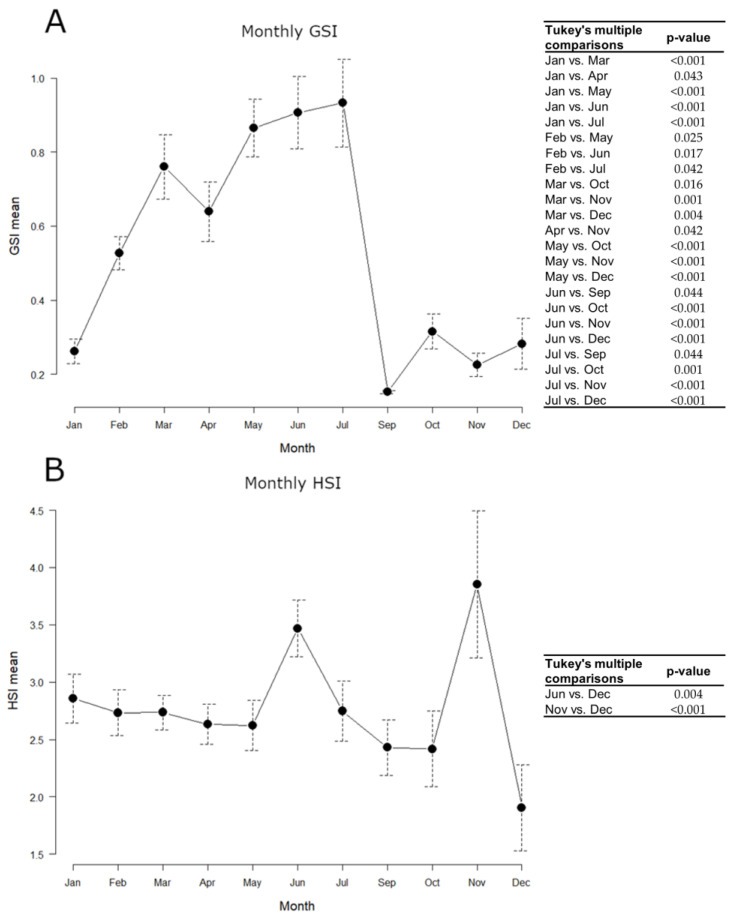
Monthly variations of European hake somatic indices throughout 2018. N = 215. (**A**) GSI mean; (**B**) HSI mean. Data are reported as *mean ± standard error of the mean (SEM)*. Tables alongside each graph indicate statistically significant differences between months evaluated by the one-way ANOVA and the post hoc Tukey’s test.

**Figure 5 biology-12-00562-f005:**
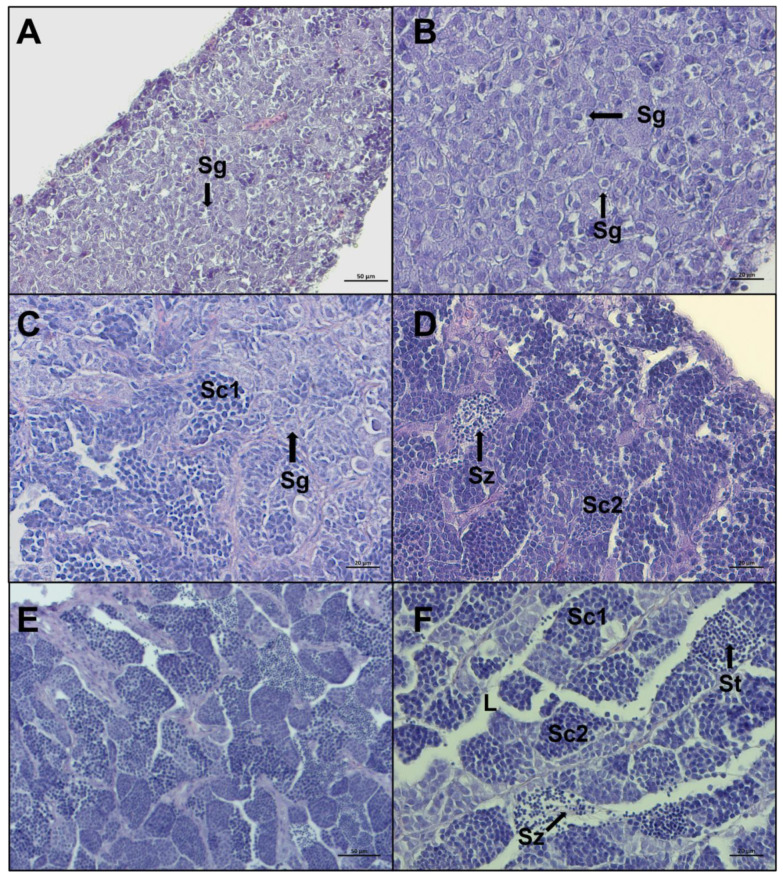
Representative histological photomicrographs of European hake testis at the first three stages of development: (**A**,**B**) *immature* stage with compact cavity-free and homogeneous tissue, only spermatogonia are present; (**C**,**D**) *developing* stage spermatocysts containing spermatogonia (Sg), primary spermatocytes (Sc1), secondary spermatocytes (Sc2), and spermatozoa (Sz), still not released in lumen; (**E**,**F**) *early spermiogenesis* stage showing almost empty lumen (L), the dominance of spermatocytes and spermatids (St), and small clusters of spermatozoa starting to be released, with evident tails. Scale bar: 50 μm for (**A**,**E**); 20 μm for (**B**–**D**,**F**).

**Figure 6 biology-12-00562-f006:**
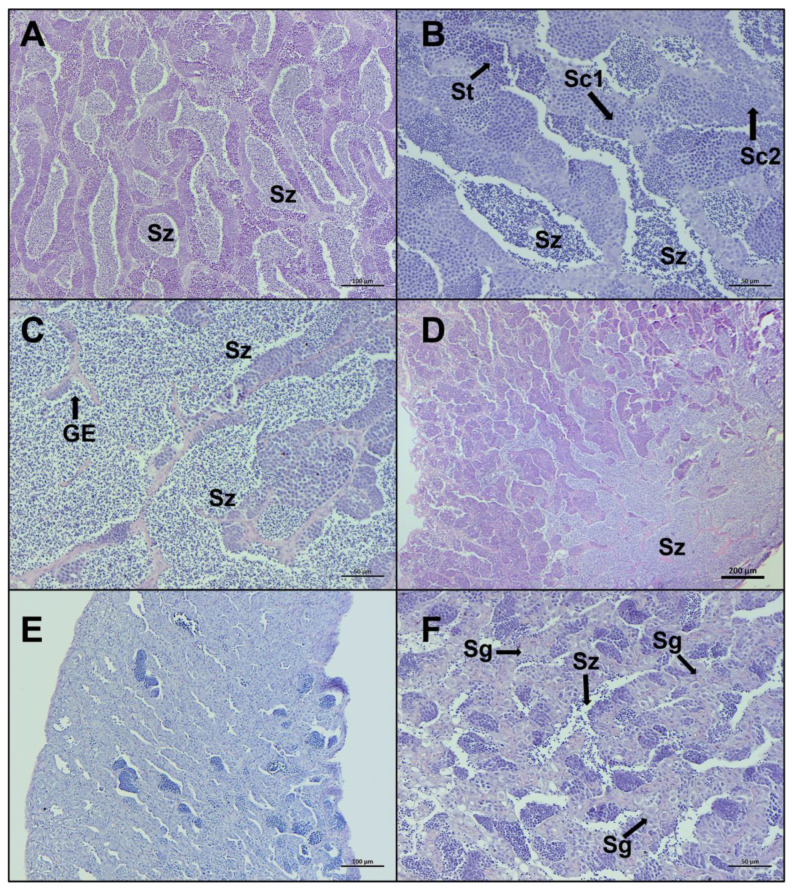
Representative histological photomicrographs of European hake testis at the last two stages of development: (**A**–**C**) *late spermiogenesis* stage with the lumen of lobules totally full of spermatozoa (Sz), discontinuous germinative epithelium (GE), and spermatocysts containing spermatocytes (Sc1, Sc2) and spermatids (St); (**D**) *late spermiogenesis* showing the confluence of spermatozoa towards the sperm duct; (**E**,**F**) *regenerating* stage when the reproductive cycle is just ended, mainly composed by spermatogonia (Sg) proliferating throughout the testis and residual spermatozoa (Sz) in the lumen. Scale bar: 200 μm for (**D**); 100 μm for (**A**,**E**); 50 μm for (**B**,**C**,**F**).

**Figure 7 biology-12-00562-f007:**
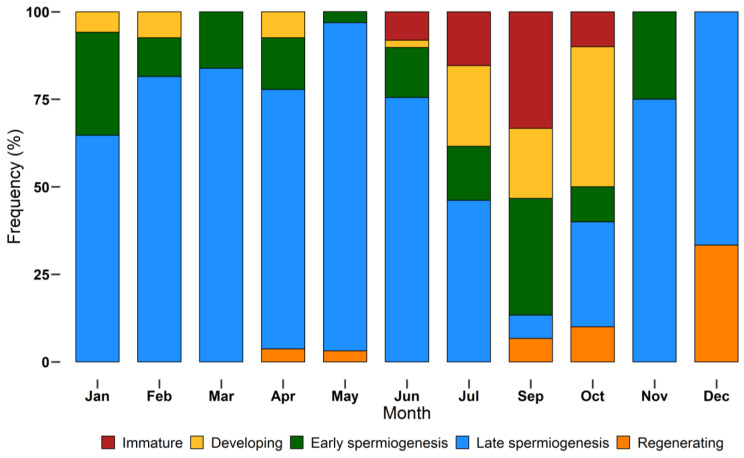
Monthly incidence of different European hake maturity stages (*immature*, *developing*, *early spermiogenesis*, *late spermiogenesis*, *regenerating*) assigned using a histological approach, during the year 2018. N = 232. Bars show the relative frequency (%) of each stage by month.

**Figure 8 biology-12-00562-f008:**
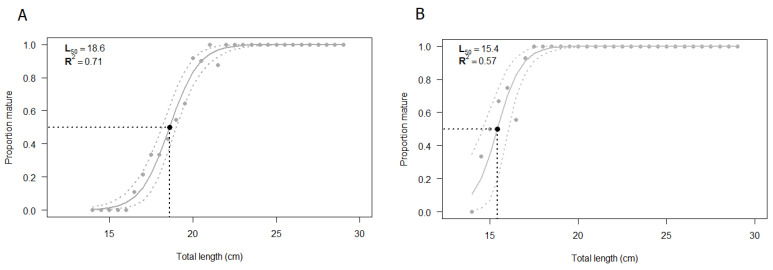
Estimated size at first maturity (L_50_) according to (**A**) macroscopic classification and (**B**) histological classification of European hake male specimens. The grey dashed lines correspond to the 95% confidence interval. N = 232. The estimation of L_50_ was performed using R statistical software version 3.6.2 (R Core Team, 2020).

**Figure 9 biology-12-00562-f009:**
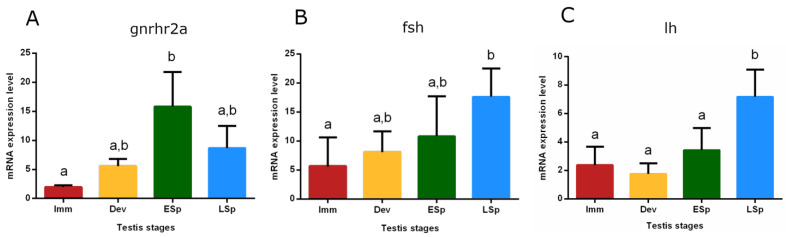
Relative mRNA expression levels of (**A**) *gnrhr2a*, (**B**) *fsh*, and (**C**) *lh* in the pituitary gland of European hake specimens at different gonadal stages (N = 20). The abundance of *gnrhr2a*, *fsh,* and *lh* transcripts was determined by qRT-PCR and normalized with *b-actin* and *18S*. Letters represent statistical significance (*p* < 0.05) among different maturity stages, as indicated by the one-way ANOVA and the post hoc Tukey’s test, performed using Prism 6 (GraphPad Software, San Diego, CA, USA). The values are *mean ± standard deviation* (*s.d.*). *Imm* = *immature* stage (N = 5); *Dev* = *developing* stage (N = 3); *ESp* = *early spermiogenesis* stage (N = 5); *LSp* = *late spermiogenesis* stage (N = 7).

**Figure 10 biology-12-00562-f010:**
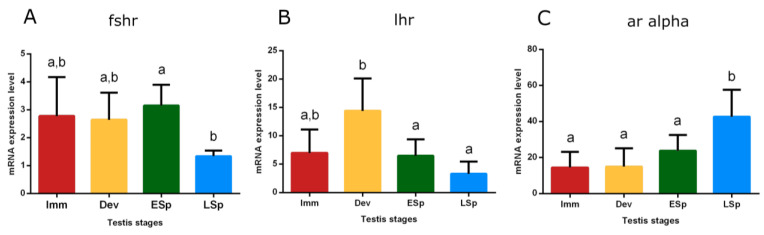
Relative mRNA expression levels of (**A**) *fshr*, (**B**) *lhr,* and (**C**) *ar alpha* in the testis of European hake specimens at different gonadal stages (N = 30). The abundance of *fshr*, *lhr,* and *ar alpha* transcripts was determined by qRT-PCR and normalized with *b-actin* and *18S*. Letters represent statistical significance (*p* < 0.05) among different maturity stages, as indicated by one-way ANOVA and post hoc Tukey’s test, performed using Prism 6 (GraphPad Software, San Diego, CA, USA). The values are *mean ± standard deviation* (*s.d*.). *Imm* = *immature* stage (N = 5); *Dev* = *developing* stage (N = 5); *ESp* = *early spermiogenesis* stage (N = 7); *LSp* = *late spermiogenesis* stage (N = 13).

**Figure 11 biology-12-00562-f011:**
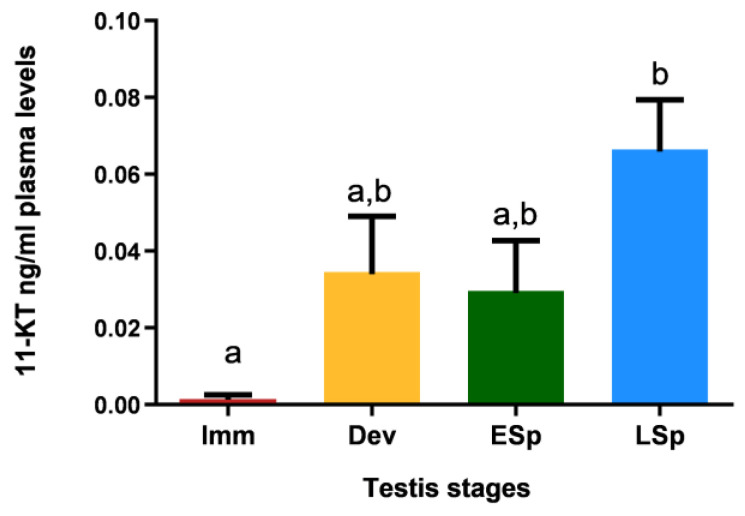
Levels of 11-ketotestosterone in plasma of European hake specimens at different gonadal stages (N = 19). Letters represent statistical significance (*p* < 0.05) among different maturity stages, as indicated by one-way ANOVA and post hoc Tukey’s test, performed using Prism 6 (GraphPad Software, San Diego, CA, USA). The values are *mean ± standard deviation (s.d*.). *Imm* = *immature* stage (N = 3); *Dev* = *developing* stage (N = 4); *ESp* = *early spermiogenesis* stage (N = 4); *LSp* = *late spermiogenesis* stage (N = 8).

**Table 1 biology-12-00562-t001:** Analysed genes and primer sequences used for real-time quantitative PCR (qPCR).

	Gene Name	Accession Number	Primer Sequence (5′-3′)	Orientation	Annealing Temperature (°C)
**Testis**	** *b-act* **	EU022566	5′-GTCATGGACTCCGGTGATGG-3′	forward	60 °C
			5′-GAGGTAGTCTGTGAGGTCGC-3′	reverse	60 °C
	** *18S* **	KF986702	5′-GAGGCCCTGTAATTGGAATG-3′	forward	60 °C
			5′-CGCAAGACACTCAACCAAGA-3′	reverse	60 °C
	** *fshr* **	KY178270	5′-CATGGCCGTGCTCATCTTC-3′	forward	58 °C
			5′-ATGAAGAGGAAGGGGTTGGC-3′	reverse	58 °C
	** *lhr* **	KY178271	5′-GTCAGCGAGTTGGACATGGA-3′	forward	61 °C
			5′-ATGACCCAGGTGAGAAAGCG-3′	reverse	61 °C
	** *ar-alpha* **	ON736432	5′-AAGCCATACCAGGTTTCCGT-3′	forward	57 °C
			5′-GATCAGGTCTGGAGCGAAGT-3′	reverse	57 °C
**Pituitary**	** *b-act* **	EU022566	5′-GTCATGGACTCCGGTGATGG-3′	forward	60 °C
**gland**			5′-GAGGTAGTCTGTGAGGTCGC-3′	reverse	60 °C
	** *18S* **	KF986702	5′-GAGGCCCTGTAATTGGAATG-3′	forward	60 °C
			5′-CGCAAGACACTCAACCAAGA-3′	reverse	60 °C
	** *fshb* **	KX377614	5′-TCTGTCGCCCAGTCAACTTC-3′	forward	58 °C
			5′-CCCACCGGACAGTCTTCAAA-3′	reverse	58 °C
	** *lhb* **	KX377615	5′-CAGCGGACACTGCATCAC-3′	forward	60 °C
			5′-ACAGTCCGGCAGCTCAAA-3′	reverse	60 °C
	** *gnrh-r2a* **	ON736433	5′-CGTTCCTCAGTTGTTCCTCT-3′	forward	60 °C
			5′-CCAGTGGGTGTCGAAGCTG-3′	reverse	60 °C

**Table 2 biology-12-00562-t002:** Summary of the Chi-square test of goodness of fit performed on the monthly sex ratio throughout the three-year period of 2017–2019. Asterisks indicate a significant difference (*p* < 0.05) in the sex ratio compared to the expected value.

MONTH	N° of Fish	N° of Males	N° of Females	Sex Ratio (M:F)	Sex Ratio (Females/Males + Females)	χ^2^	*p*-Value
January	122	37	85	0.44:1	0.697	18.885	1.388 × 10^−5^ *
February	123	64	59	1.08:1	0.480	0.20325	0.6521
March	122	69	53	1.30:1	0.434	2.0984	0.1475
April	133	79	54	1.46:1	0.406	4.6992	0.03018 *
May	142	78	64	1.22:1	0.451	1.3803	0.2401
June	113	70	43	1.63:1	0.381	6.4513	0.01109 *
July	95	38	57	0.67:1	0.600	3.80	0.05125
September	105	53	52	1.02:1	0.495	0.009524	0.9223
October	111	48	63	0.76:1	0.568	2.027	0.1545
November	117	26	91	0.29:1	0.778	36.111	1.864 × 10^−9^ *
December	133	36	97	0.37:1	0.729	27.977	1.227 × 10^−7^ *
**TOTAL**	**1316**	**598**	**718**	**0.83:1**	**0.545**	**10.942**	**0.00094 ***

**Table 3 biology-12-00562-t003:** Summary of the size at first maturity (L_50_) estimated based on macroscopic and histological classifications of gonadal maturity stages of European hake males. Logit regression and the Wald Test were performed in the R environment (R Core Team, 2020).

L_50_ MACROSCOPIC				
*Coefficient*	*Estimate*	*Std. Error (SE)*	*z Value*	*p Value*
a	−21.08	3.14	−6.73	<0.0001
b	1.13	0.17	6.81	<0.0001
**L_50_ HISTOLOGICAL**				
** *Coefficient* **	** *Estimate* **	** *Std. Error (SE)* **	** *z Value* **	** *p Value* **
a	−22.13	6.02	−3.68	0.00023
b	1.44	0.37	3.89	0.00010

**Table 4 biology-12-00562-t004:** Summary of L_50_ values and comparison between macroscopic and histological methods. Likelihood ratio test and Cohen’s k coefficient of the agreement were performed in the R environment (R Core Team, 2020).

	Estimation of Size at First Maturity			Likelihood Ratio Test			Cohen’s k Coefficient of Agreement				
	*Value*	*Conf. Interval*	*R^2^_MF_*	*LogLik*	*Chi-sq*	*p Value*	*Value*	*Conf. Interval*	*ASE*	*z Value*	*p Value*
**Macroscopic** **Method**	18.6	18.2–19.0	0.71	−59.42	73.63	<0.0001	0.24	0.13–0.36	0.06	4.10	0.000042
**Histological** **Method**	15.4	14.6–16.1	0.57	−22.61							

## Data Availability

Data available on request due to restrictions, e.g., privacy or ethical.

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
