# Peer review of "Reproductive Biology of Male European Hake (Merluccius merluccius) in Central Mediterranean Sea: An Overview from Macroscopic to Molecular Investigation"

_biology, 2023, doi:10.3390/biology12040562_

Round 1

Reviewer 2 Report

Please find the attached manuscript with the reviewer’s detailed corrections.

There is a need of some improvements. 

The manuscript is not well written and in some paragraphs it is difficult to understand what the authors mean. Also some changes had to be done in Materials and Methods. Reconsider sex ratio estimation.

The revised manuscript has to be proofreading, since the English in the present manuscript is not of publication quality and requires some improvement. Please carefully proof-read spell check to eliminate grammatical errors.

Round 2

Reviewer 1 Report

The changes made by the authors have improved the formal quality of the document: English language, omissions or errors on cited bibliographic references, and they proposed an improved version of Figure 4.
My preference is the 4ter proposal, which also allows authors to give the most complete information possible for the reader, concerning significant inter-sampling differences.  
I thank the authors for responding to all the criticisms I had raised.
My opinion is now that the document can be published in this new version.